# Enhanced Strength and Hardness of AS41 Magnesium Alloy Fabricated by Selective Laser Melting

**DOI:** 10.3390/ma15175863

**Published:** 2022-08-25

**Authors:** Ruirui Yang, Keyu Chen, Shifeng Wen, Shijie Zhu, Haotian Qin, Xiaochao Wu, Yan Zhou, Yusi Che, Yusheng Shi, Jilin He

**Affiliations:** 1Zhongyuan Critical Metals Laboratory, Zhengzhou 450001, China; 2School of Materials Science and Engineering, Zhengzhou University, Zhengzhou 450001, China; 3State Key Laboratory of Material Processing and Die & Mold Technology, School of Materials Science and Engineering, Huazhong University of Science and Technology, Wuhan 430074, China

**Keywords:** AS41 magnesium alloy, selective laser melting, laser power density, high-angle grain boundaries, strength

## Abstract

AS41 magnesium alloy possesses outstanding performance features such as light weight, high strength to toughness ratio and excellent heat resistance due to the addition of Si element, while traditional casting methods are prone to inducing large grain size and coarse Mg_2_Si phase. In this study, we first reported utilizing the selective laser melting (SLM) technique, fabricating AS41 samples and exploring the effect of laser energy densities on the metallurgical quality by characterizing and investigating the microstructure and mechanical properties. Results showed that the optimal laser energy density range was 60 to 100 J/mm^3^. Average grain size of only 2.9 μm was obtained with weak texture strength of 1.65 in {0001} orientation. Meanwhile, many dispersed secondary β-Mg_17_Al_12_ and Mg_2_Si phases were distributed inside the α-Mg matrix. It was confirmed that the SLM process introduced more grain recrystallization, inducing giant high-angle grain boundaries (HAGBs) and hindering the movement of dislocations, therefore forming dislocation strengthening while achieving grain refinement strengthening. Finally, three times the ultimate tensile strength of 313.7 MPa and higher microhardness of 96.4 HV than those of the as-cast state were obtained, verifying that the combined effect of grain refinement, solid solution strengthening and precipitation strengthening was responsible for the increased strength. This work provides new insight and a new approach to preparing AS41 magnesium alloy.

## 1. Introduction

Magnesium (Mg) alloy has the vast advantages of high specific strength and stiffness, firm damping and electronic shielding ability, and degradability [1,2]; it is widely used in automotive, biomedical, aerospace and other fields. In recent years, Mg alloy parts have been mainly formed by casting technology [3,4,5,6,7,8], which has two obvious limitations. On the one hand, the prepared Mg alloys inevitably produce casting defects, such as shrinkage porosity, inclusions and segregation [9,10]. On the other hand, as-cast Mg alloys rely on molds to form, making it difficult or even impossible to manufacture parts with complex shapes and structures. The above two factors limit the in-depth development of performance and the wide application of Mg alloy. Therefore, a novel manufacturing method is urgently needed to achieve complex construction and high-quality forming of Mg alloy.

Selective laser melting (SLM), as one of the mainstream technologies of metal additive manufacturing, can manufacture almost entirely dense metal parts with complex geometries and precise structures [11] without the need for expensive molds, which can significantly shorten the production cycle. More importantly, an appreciably fine-grained microstructure is formed due to the rapid melting and solidification process, which significantly improves the mechanical properties of the as-acquired parts. For this reason, researchers have carried out a series of studies on the feasibility and performance analysis of SLM Mg alloys.

For instance, C. C. Ng et al. [12] reported the microstructure and mechanical properties of pure Mg formed by SLM, and the study showed that the grain size of pure Mg was in the range of 2–5 μm with hardness of 0.59–0.95 GPa and elastic modulus of 27–33 GPa, which matched well with the mechanical properties of human bones. Hu et al. [13] studied the effects of powder particle sizes and scanning time intervals on the surface roughness and microhardness of as-built bulk pure Mg samples by SLM. Results showed the relative density reached more than 95% with no obvious spheroidization on the sample surface and microhardness of 52 HV, which was much higher than that of as-cast pure Mg. Compared with pure Mg, the strength and plasticity of Mg-Al type Mg alloys are significantly improved due to the addition of aluminum (Al). Niu et al. [14] reported that the relative density of Mg alloy parts with 9 wt.% Al added by SLM was 95.7%, and the ultimate tensile strength (UTS) was as high as 274 MPa. Zhang et al. [15] also reported an SLM study based on Mg-Al alloys with the same content, and the average microhardness of the samples could reach an unprecedented 75 HV. Not limited to this, trace amounts of zinc (Zn) have been added to the Mg-Al series Mg alloys to produce new Mg-Al-Zn alloys (AZ-type) to further improve strength and corrosion resistance [1]. At present, SLM research on AZ series Mg alloys has mainly focused on AZ91D and AZ31B types [16,17,18,19,20,21].

AS41 Mg alloy, a kind of heat-resistant Mg alloy compounded by adding Al and silicon (Si) elements with metal Mg as the matrix, is a typical representation of AS-grade Mg alloy. Due to the addition of Si [1], AS41 Mg alloy has good heat resistance and high creep resistance, and is generally considered to possess better overall performance than AZ91 alloy [22]. However, as-cast AS41 alloy often has reticulated Mg_17_Al_12_ and coarse Mg_2_Si phases and its mechanical properties are poor at room temperature. To overcome this problem, researchers tried to control the size and improve the morphology of Mg_2_Si by employing processing technologies such as rapid solidification and hot extrusion [22,23] or by adding various alloying elements [24,25,26]. Recently, SLM technique, as the rapid melting and rapid solidification characteristics, has been considered to have the potential to eliminate the coarser Mg_2_Si phase to improve the mechanical properties of AS41 Mg alloys. For instance, Wang et al. [27] fabricated the Mg-Y-Sm-Zn-Zr alloy with increased hardness by SLM. The precipitation of Mg12(Y, Sm) Zn phase was restrained due to a higher fast cooling rate, resulting in more eutectic (Mg, Zn)3(Y, Sm) phase and new Y_2_O_3_ phase with higher hardness formed. Wei et al. [18] found that SLM methods had an influence on phase. The SLMed sample that contained β precipitates existed in the form of totally divorced eutectic structure. However, through the die-cast process, it could be found that some bulky divorced eutectic β-Mg17Al12 was surrounded by the lamellar eutectic structure. Wu et al. [28] also pointed out that the phase transition process was inhibited and the precipitated phase was changed when ZK60 was prepared by SLM process.

In this work, the AS41 Mg alloy fabricated by the SLM process was investigated for the first time. The processing feasibility and the effect of laser energy densities on the AS41 Mg alloy were discussed; we aimed to obtain AS41 Mg alloy samples with an accurate process window and excellent mechanical properties. The microstructure and mechanical properties of the AS41 Mg alloy formed under the optimal process parameters were observed and measured, and the interaction between the process, microstructure and mechanical properties was elucidated. Finally, the comprehensive mechanism of SLM-formed AS41 Mg alloy was deeply explored to provide theoretical and technical references for the preparation of high-performance Mg alloy.

## 2. Materials and Methods

Gas-atomized AS41 powders with particle sizes ranging from 15 μm to 53 μm and a theoretical density of 1.79 g/cm^3^ were used as the raw materials, which were provided by Zhongkeyannuo New Material Science and Technology Co., Ltd. The morphologies are shown in Figure 1a. The chemical compositions of AS41 powders used in this study are listed in Table 1. The HK M300 SLM machine (HUAKE 3D, Wuhan, China) was employed to fabricate the AS41 sample. Before the manufacturing process, the building chamber of the SLM machine was filled with argon to achieve oxygen content below 100 ppm. Figure 1b schematically shows the bidirectional scanning strategy. The laser beam moved in a zigzag way across the surface, adjacent scan tracks developed along with two opposite directions, and the rotation angle of the laser between layers was 67°.

The metallographic and electron microscopy testing was needed for the sample preparation process, including grinding, polishing and corrosion. Firstly, the grinding process used 400–800# abrasive papers and 1000–2000# waterproof abrasive papers. Next, 0.25 μm polishing agent was used to polish samples. After polishing, ethanol was used to carry out ultrasonic cleaning. The mirror surface of the sample was immersed in the etching agent for 2 s to corrode samples. Finally, the sample was immediately washed with water for 30–60 s and anhydrous ethanol for 20–30 s. X-ray diffraction (XRD) (X’pert3 Powder, the Netherlands) with Cu K-alpha radiation (λ = 0.15418 nm) was applied to determine the phase constitution for the SLM samples perpendicular to the building direction. The metallograph of the SLM samples was observed by standard techniques using optical microscopy (Leica DM400). Microstructures of the samples were characterized by scanning electron microscopy (Zeiss/Auriga-bu) and transmission electron microscopy (FEI Tecnai G2 F30). The electron backscattered diffraction (EBSD) test samples were electrolytically polished on LectroPol-5 (Struers, Ballerup, Denmark). EBSD was performed on the orientation imaging microscope system (Oxford Instruments, Oxford, UK) mounted on the JSM-7600F (JEOL, Tokyo, Japan) scanning electron microscope. The polished sample was prepared for Vickers microhardness testing (HV-1000A) under a maximum load of 50 g and a dwell time of 10 s at room temperature. The dog-bone shape specimens (Figure 2) were fabricated for mechanical tests according to the GB/T 16865-2013 standard.

## 3. Results and Discussion

### 3.1. Processing Map

As shown in Figure 3, the processing map displays the AS41 samples formed under different laser powers and scanning speeds. To evaluate the combined effect of the above parameters and control the SLM process integrally, one single factor termed “volume energy density” (Ev) was defined as follows:(1)$$Ev=\frac{P}{HsTV}$$
where *P* is the laser power, *V* is the scanning speed, *Hs* is the scanning interval and *T* is the interlayer thickness, respectively, and the unit of Ev is J/mm^3^.

According to the value of Ev and the effect of formability of SLM AS41, the processing map could be divided into A, B and C Zones, as shown in Figure 3a. Several samples were formed for each zone, one of which is shown in the surrounding image as Figure 3b–f. The quality of forming can be judged by observing the surface pores of samples in metallographic photographs [29,30]. For a given Ev below ~65 J/mm^3^ in Zone A, there was a mass of pore defects in the metallographic microscope specimen, as shown in Figure 3b,c. Meanwhile, some powder could not be fully melted due to insufficient energy, resulting in the balling phenomenon. With the volume energy density of Ev increasing, the number of pore defects in Zone B gradually decreased, as shown in Figure 3d. Furthermore, the balling phenomenon was hardly observed as the Ev ranged between 65 J/mm^3^ and 100 J/mm^3^. As the energy density continued to increase to above ~100 J/mm^3^ in Zone C, the forming quality began to deteriorate, as shown in Figure 3e,f.

The influence of the same Ev with different scanning speeds on the molten pool morphology was also investigated, as shown in Figure 3g. Although the Ev was the same, the molten pool morphology was obviously different in shape and size. When the scanning speed was higher, the scanning trace would be tiny, resulting in a decrease in overlap area, as shown at the bottom of Figure 3g. On the other hand, the overlap of the molten pool became broad as the scanning speed decreased at the top of Figure 3g. Zhou et al. [31] also pointed out that if the scanning speed was too high, the molten pool was discontinuous; when the laser power and the scanning speed became lower, the molten pool became stable, and the track was continuous and clear. Nevertheless, the laser power cannot be too low because insufficient laser energy can cause the balling phenomenon.

### 3.2. Phase and Microstructure

XRD patterns of the SLM samples at different laser energy densities are shown in Figure 4. The samples were composed of α-Mg, β-Mg_17_Al_12_ and Mg_2_Si phases at each *E_V_*. Furthermore, the effect of different Ev on the phase was slight; the angle and intensity of the feature peaks remained similar when the Ev increased from 48 J/mm^3^ to 126 J/mm^3^. However, the α-Mg peaks of the SLM sample slightly shifted to high diffraction angles compared with the standard PDF card of Mg (PDF#89-5003). According to Bragg’s law, the lattice constant (*d*) decreases while the diffraction angle increases. There were two reasons why *d* decreased in this work, one from the microstructure and the other from the lattice. Generally, the grain size was small, owing to the rapid solidification during the SLM process. In addition, the elements of Al, Si and Zn acted as substitution solutes in the Mg matrix based on solid solution theory. Moreover, the atomic radius of all three elements was less than the atomic radius of Mg. Therefore, the *d* decreased with the solid solubility increasing due to SLM technology.

.

The metallographic microscope photograph of samples at different Ev is displayed in Figure 5. The metallographic microscope photograph of the cross-sectional sample with laser energy density at 80 J/mm^3^ is shown in Figure 5a,b. There were closely striated scanning trajectories, and Figure 5a clearly shows a closely striated scanning trajectory, and in Figure 5b, there is an overlapping region (OLR) between the molten pools during the scanning process. The grain size in the sample is slightly larger than the center of the tracks (CST) due to the reheating process. Meanwhile, overlapping regions make the sample bonding better because of the existence of overlapping regions. Figure 5c,d show the longitudinal section track of the sample, and the tracks of Figure 5c,d are different. When Ev is 65 J/mm^3^ in Figure 5d, the bottom of the sample appears elliptical by the Gaussian distribution of laser energy aligned layer by layer. However, in Figure 5c, when the Ev is 80 J/mm^3^, the width of the molten pool increases with the increase of *E_V_*. The ratio of width to height of the molten pool causes the bottom elliptical feature to be covered, and the longitudinal elliptical track tends to be flat. Figure 3g is a schematic diagram of a track forming a flat vertical section. When the laser energy density increases, the width-height ratio of the molten pool continues to change, and the layer-like elliptical features become prominent again. At the same time, the depth of the molten pool is several times the thickness of the layer, and each layer will be reheated several times during the forming process [18]. This forming process allows for a good metallurgical bond between adjacent tracks and layers.

Figure 6 shows the EBSD analysis in the cross-section of the AS42 alloy specimen with the grain orientation map, {0001} pole figure and grain size distribution. The grain orientation map in Figure 6a provides an insight into the microstructural features of the sample. Moreover, it shows that the sample had no obvious preferred orientation. It can easily be found that the areas of the three colors basically remained the same, indicating that the grains at (0001), ($\bar{1}$2$\bar{1}$0) and (01$\bar{1}$0) orientations equally existed on the cross-section. This can also be confirmed from the {0001} pole figure (Figure 6b): the sample exhibits a weak texture in the {0001} direction with maximum texture strength of 1.65. At the same time, there were prominent equiaxed grains in the whole view, about 100 μm × 100 μm in Figure 6a. The width of the single molten track was approximately 150 μm, according to Figure 5a. The homogeneous and refined grains were obtained owing to almost the same thermodynamics of solidification in this area, as shown in the orientation image map. Furthermore, from the statistical data obtained by using HKL Channel 5 analysis software, the average grain size of the SLM AS41 sample was 2.9 µm (Figure 6c). In fact, the grain size is determined by the thermal gradient (G), the solidification rate (R) and the cooling rate (T = G × R). The grain size decreased with the increase in the cooling rate of the solidifying material during the SLM process. The ultrahigh cooling rate (10^3^–10^8^ K s^−1^) of SLM generates a large temperature gradient within a melt pool and provides a shorter growth time for primary grains and secondary grown regenerated sub-grains. Moreover, a giant temperature gradient causes a surface tension gradient and resultant thermal convection, facilitating the grown columnar dendrites to remelt and transform into refined secondary grains, resulting in a relatively smaller average grain size.

Figure 7 shows the EBSD analysis of the SLM AS41 sample, which included the crystal orientation, dislocation density and residual plastic strain, recrystallization distribution and Schmid factor value distribution. Figure 7a illustrates the spatial distribution of the average kernel misorientation in the as-built AS41 alloy sample by the SLM technique. KAM is defined as the average orientation difference between each measurement point and its nearest neighbor, with orientation differences exceeding a critical value of 5° excluded from the calculation; this is usually used for evaluating the endogenous dislocation density and its spatial distribution. A higher value of KAM indicates a greater degree of plastic deformation or a higher dislocation density. It could be found from Figure 7b that a higher volume of dislocations distributed evenly surrounded the grain boundaries, as indicated by a lighter green color. The dislocations aggregated and pinned at the grain boundary are considered to be the source of the dislocation strengthening effect and can effectively hinder the further growth of grains and play the role of grain refinement strengthening. Dislocation aggregations and entanglements can also be observed in Figure 9f. Figure 7b shows the grain boundary misorientation angle distribution obtained from the top surface of the AS41 sample. Generally, grain boundaries with a misorientation angle greater than 15° are defined as high-angle grain boundaries (HAGBs). In comparison, grain boundaries with a misorientation angle of less than 15° are defined as low-angle grain boundaries (LAGBs). Boundaries are color-coded with green LAGBs and black HAGBs. Figure 7f provides a more intuitive regular distribution pattern of the grain boundary misorientation angles. The linear fraction of HAGBs in this region was calculated to be 90.2%. The formation of HAGBs is mainly related to the dynamic recrystallization process that occurred in the samples. The recrystallization process forms new strain-free grains through the formation and causes subsequent movement of new mobile high-angle grain boundaries. Figure 7d,h also confirm that a large number of recrystallization processes took place inside the samples. In general, the strength and toughness of material are closely related to the proportion of HAGBs in its structure. The contribution of HAGBs and LAGBs to the strength is mainly reflected in the strength of the hindering effect on the dislocation movement. The LAGBs with the boundary orientation below a certain critical angle easily allow dislocations to pass through the interface, while the HAGBs hinder the further movement of dislocations, resulting in dislocation pinning and aggregation at the grain boundaries. HAGBs can hinder crack propagation; that is, the higher the HAGBs fraction, the better the strength and toughness of the material. In addition, the larger the misorientation angle of adjacent grain boundaries, the stronger the ability of the material to resist crack propagation. Therefore, it can be predicted that the SLM-formed AS41 samples can achieve higher strength and toughness values (compared to the as-cast samples), considering only the effect of grain boundary misorientation.

Figure 7c depicts the Schmid factor distribution of the selected region in the AS41 alloy sample. The formula for calculating the Schmid factor is:(2)$$\sigma_{s}=\tau_{s}\frac{1}{\cos\lambda \cos\varphi }$$
where *λ* is the angle between the applied load *F* and the {110} orientation, *φ* is the angle between the applied load *F* and the normal direction of {111}, and *τ_s_* is the critical shear stress. The product of cos *λ* and cos *φ* is called the Schmid factor. The plastic deformation of the material is mainly completed by slip, and the higher the Schmid factor, the greater the probability of the slip system being activated. It can be seen that the Schmid factor is primarily distributed in the range of 0.29–0.48. The average Schmid factor value of the obtained region is calculated to be 0.32, which means there is a relatively large tendency toward plastic deformation when applying the external load to the alloy sample. Figure 7d shows the overall recrystallization distribution of the selected sample defined by three types of grains. The blue areas correspond to the recrystallization grains, with an internal orientation difference of less than 2°; the red areas correspond to the deformed grains with an orientation difference ranging from 2° to 7.5°; and the yellow areas are the sub-grains (7.5°–15°). The blue region dominates the entire test surface with the highest proportion (81.5%, as indicated in Figure 7h), which means that the as-built sample has undergone an obvious recrystallization nucleation and growth process. The recrystallization process can be attributed to the following: firstly, the extremely fast cooling and solidification rates introduce significant local strains, resulting in appreciable stored energy that accumulates in the as-built samples through repeated treatments of each layer; secondly, the inevitable phase transition in the AS41 sample results in a distorted lattice that facilitates further storage of this energy; finally, the orientation gradient of the grains can also facilitate recrystallization nucleation.

Figure 8a,b display different microstructure scales and phase morphology of the as-built AS41 sample manufactured with 80 J/mm^3^ laser energy. The phase of AS41 Mg alloy is composed of α-Mg matrix phase, β-Mg_17_Al_12_ and Mg_2_Si phase, of which β-Mg_17_Al_12_ phase and Mg_2_Si phase are the strengthening phases. As shown in Figure 8, some point-like nanoscale Mg_2_Si phase and microscale β-Mg17Al12 phase precipitates are found to be randomly distributed in the sample with no coarser Mg_2_Si phase, but β-Mg_17_Al_12_ phase precipitates exist, which is consistent with the as-cast sample. Usually, the β-Mg_17_Al_12_ phase is discontinuously distributed in the grain boundary in a coarse network in the as-cast sample, while in terms of SLM, the β-Mg_17_Al_12_ phase is dispersed in the grain boundary and within the grain in the form of fine particles. At the same time, the size of the Mg_2_Si phase decreases and partially forms a granular structure compared with the as-cast state. In the AS41 Mg alloy formed by SLM, the integrated coupling effect is caused by dispersed Mg_2_Si phase and β-Mg_17_Al_12_ phases pinning the grain boundaries, and precipitated particles hindering the movement of dislocations during stretching can theoretically increase the yield strength of samples when compared with the as-cast samples.

Figure 9a shows a STEM image of the detailed second phase distribution and morphology within the selected measurement region. It indicates herringbone-like or irregular block-like secondary phases (shown as bright grey areas) on the Mg-4Al-1Si matrix and grain boundaries. Figure 9b is the high-magnification STEM image of a specific region selected by the orange frame in Figure 9a. Figure 9c–e and Figure 10 show chemical mapping of the selected entire region of Figure 9b and energy-dispersive spectrometer (EDS) mapping of the four detection points, respectively. The bright grey herringbone-like and irregular massive regions in Figure 9b were detected as Al-rich and Si-rich segregated alloy phases. The EDS point-scanning calculation results can roughly determine the atomic ratios of the main alloying elements such as Mg, Al, Si, etc., thereby confirming that the alloy phase in the bright grey area is a composite of at least two phases. In order to quantitatively determine the secondary composition phase of the Mg alloy contained in the α-Mg matrix, the SAED pattern of the selected area was obtained. After comparison with the standard PDF card, the β-Mg_17_Al_12_ phase with a crystal belt axis of [00$\bar{8}$] was proven to exist. Hence, a comprehensive analysis carried out on these tests combining the XRD analysis can basically determine that there is secondary β-Mg_17_Al_12_ phase and Mg_2_Si phase on the matrix of the AS41 alloy sample.

From the transmission electron microscope (TEM) image of a specific region in Figure 9f, it can be found that a large number of nanoscale accumulated dislocations piled up at the grain boundaries, as shown by the irregular rod-shaped dark grey area in the figure. To analyze the type and characteristics of these dislocations, the inverse Fourier filtered transform pattern of the dislocations in Figure 9f was obtained (shown in Figure 9h), and typical edge dislocations with partial lattice distortion (shown in dark blue dashed frames) can be found. As we all know, when the dislocation moves to the grain boundary, the grain boundary hinders the movement of the dislocation. In this way, the dislocation will accumulate near the grain boundary, which will cause the strain hardening of the material so that the material has better strength and plasticity. For polycrystalline AS41 Mg alloy material, the dislocation slip is strongly influenced by numerous internal grain boundaries. The Hall–Petch relationship shows that the yield stress of polycrystalline materials is inversely proportional to the square root of the average grain size. The more grain boundaries per unit volume of a material, the stronger the material, and the lattice dislocations will not easily slip across the grain boundaries but will instead pile up behind the grain boundaries. From the EBSD analysis in Figure 6, it can be seen that the SLM process introduces grains and sub-grains with smaller average sizes, thereby introducing more grain boundaries. Therefore, the UTS of SLM-formed AS41 Mg alloy has increased compared with traditional casting and other processing methods. This will be confirmed later in the mechanical properties section.

### 3.3. Mechanical Properties

Figure 11 depicts the stress-strain curves of 12 sets of tensile specimens fabricated at the laser densities ranging from 48 J/mm^3^ to 126 J/mm^3^. The UTS of most tensile specimens exceeds 220 MPa, except the UTS of approximately 100 MPa corresponding to a laser energy density of 60 J/mm^3^, which can be attributed to unavoidable experimental deviation. The samples with laser densities in the range of 60 J/mm^3^ to 100 J/mm^3^ (Figure 11b) have the highest average UTS, attributed to the samples possessing the fewest formed metallurgical defects (as shown in Figure 3a–f). The maximum UTS value of all measured samples is 313.7 MPa when laser energy density reaches 80 J/mm^3^. A significant improvement (nearly three-fold) of UTS has been achieved on SLM AS41 alloys when compared with the as-cast AS41 Mg alloys of 107 MPa obtained by Xu et al. [24]. The enhancement mechanism of strength in as-built AS41 Mg alloys by SLM could be attributed to three main reasons: (i) fine-grain strengthening; the more refined average grain size of 2.9 μm (Figure 6c) has been achieved in this work due to the steep temperature gradient and fast solidification rate, according to the Hall–Petch formula: σ_s_ = σ_0_ + Kd^−1/2^ (σ_0_: intracrystalline resistance to deformation; K: the influence coefficient of grain boundary on deformation). A smaller grain size means more excellent resistance to deformation, i.e., greater UTS at tensile fracture. (ii) Dislocation strengthening: microstructure analysis shows that the material is mainly composed of uniformly distributed equiaxed grains without obvious grain orientation. A large number of grain boundaries are randomly distributed, and almost all of them are high-angle grain boundaries (Figure 7b), which can effectively hinder the movement of dislocations. Dislocations are easy-to-form blocking products on the high-angle grain boundaries, which significantly improve the strength of the material. (iii) Solid solution and precipitation strengthening: compared with the as-cast AS41 Mg alloy, there is a non-negligible solid-solution-strengthening effect because the rapidly solidified solute atoms cannot diffuse in time. In addition, microstructure analysis showed that the coarser primary phase was replaced by the finely dispersed secondary phase, which had an excellent precipitation-strengthening effect.

Figure 11b reflects the variation of Vickers hardness values for twelve groups of bulk specimens formed at different laser densities. It can be found that the laser energy density has little effect on the hardness value of the AS41 alloy samples, as the hardness value of all samples fluctuates in a small range from 92.9 HV to 101.2 HV, with an average value of 96.4 ± 2.73 HV. The full hardness value of metallic materials contains four parts: grain-boundary-hardening ΔHV_GB_, solid-solution-hardening ΔHV_SS_ and precipitation-hardening ΔHV_pre_, and can be calculated by the formula [27,32]: HV = ΔHV_0_ + ΔHV_GB_ + ΔHV_SS_ + ΔHV_pre_, where ΔHV_0_ is the hardness of undeformed pure Mg (6.0 HV). The hardness of grain boundary hardening (ΔHV_GB_) is calculated by the Hall–Petch formula: HV = H_0_ + Kd^−1/2^ (H_0_ = 51 HV, k = 0.022/m^−1/2^). The average grain size of the as-built AS41 sample is calculated to be 2.9 μm, and this provides ΔHV_GB_ = 64 HV. Considering that Si is basically insoluble with the Mg matrix (α-Mg), and the Mn content is extremely low (0.19 wt.% in raw powders), only the solid-solution-strengthening effect of Al and Zn elements is considered in this work. Therefore, solid solution hardening (ΔHVss) can be expressed by: ΔHVss = C (k_Al_^1/n^ c_Al_ + k_Zn_^1/n^ c_Zn_)^n^, where *C* is defined as a constant of about ~0.3, *n* is taken as 1/2, and *k_Al_*, *k_Zn_* are solid solution factors of solute elements 118 MPa (at.%)^−1/2^ and 578 MPa (at.%)^−1/2^, respectively. *c_Al_* and *c_Zn_* are the atomic fraction content of Al and Zn elements, respectively. The solid-solution-hardening value (ΔHVss) was calculated to be about 8.5 HV. The theoretical microhardness of the as-deposited AS41 alloy sample summed by grain-boundary-hardening ΔHV_GB_ (64 HV), solid-solution-hardening ΔHV_SS_ (8.5 HV) and ΔHV_0_ (6 HV) was calculated to be 78.5 HV, which is somewhat lower than the actual value (92.9–101.2 HV). Hence, there is an inevitable precipitation-strengthening (ΔHV_pre_) effect inside the AS41 alloy matrix, which is related to the precipitation hardening of the dispersed β-Mg_17_Al_12_ and Mg_2_Si phases. It is concluded that the mechanism of hardness strengthening of AS41 Mg alloy formed by SLM mainly results from grain refinement, solid solution strengthening and precipitation hardening.

Figure 12 shows typical fracture morphologies of the tensile specimen fabricated at a laser energy density of 80 J/mm^3^. Figure 12a illustrates the tensile sample exhibiting a ductile-brittle hybrid fracture with a dominant brittle fracture. The high magnification electron microscope image (shown in Figure 12b) of the specific region in Figure 11a shows obvious quasi-dissociative tearing ridges (as marked by red circles in Figure 12b) and torn dimples (as marked by yellow circles in Figure 12b). The ductile fracture feature is contributed to the α-Mg matrix with toughness characteristics, while brittleness is mainly related to the existence of brittle β-Mg_17_Al_12_ and Mg_2_Si phases distributed randomly among the α-Mg matrix. Moreover, some holes, cracks and impurity doping were found; these metallurgical defects are unavoidable for the SLM process. The forming quality can be further improved by improving other process parameters besides laser power and scanning speed.

## 4. Conclusions

In this study, the high-performance AS41 alloy was fabricated by SLM. The laser energy density parameters were optimized and the microstructure and mechanical properties of the SLMed samples were characterized. Further, the strengthening mechanism of the SLMed samples was discussed systematically. The main conclusions can be summarized as follows: (1) the optimum laser energy density range for forming AS41 alloy was determined to be 65–100 J/mm^3^; (2) the AS41 alloy samples formed by SLM possessed a smaller average grain size of 2.9 μm than the as-cast alloys and they exhibited weak texture and fine-grain strengthening effects. Secondary β-Mg_17_Al_12_ and Mg_2_Si phases uniformly dispersed in the α-Mg matrix caused the accumulation and pinning of a large number of dislocations at the grain boundary, which resulted in a considerable dislocation-strengthening effect; (3) a significant improvement (nearly three-fold) of UTS (313.7 MPa) was achieved on SLM AS41 alloys compared with the as-cast AS41 alloys, ascribed to the combined action of fine-grain strengthening and solid solution strengthening. A relatively high average microhardness value of 96.4 HV was achieved, attributed to the synergy of grain refinement, solid solution strengthening and precipitation hardening.

## Figures and Tables

**Figure 1 materials-15-05863-f001:**
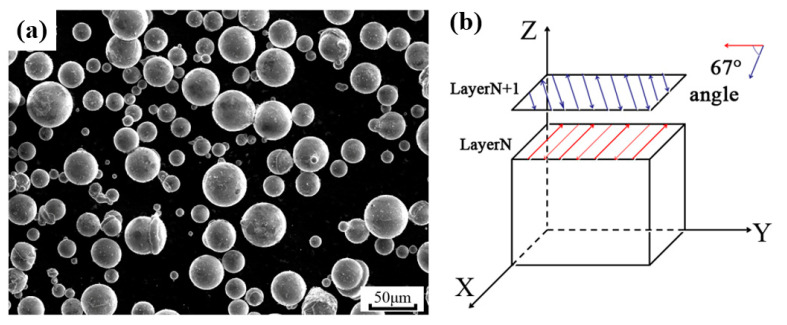
(**a**) SEM morphology of AS41 powders, (**b**) the laser scanning strategy with a hatch rotation of 67°of SLM process.

**Figure 2 materials-15-05863-f002:**
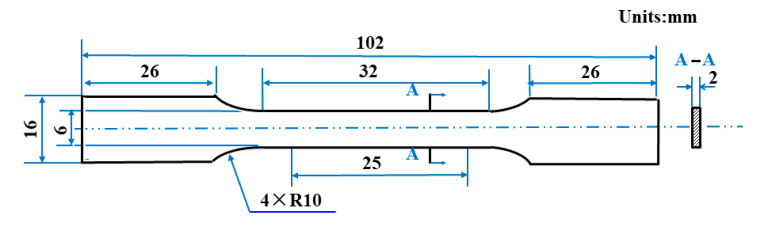
Diagram of tensile sample.

**Figure 3 materials-15-05863-f003:**
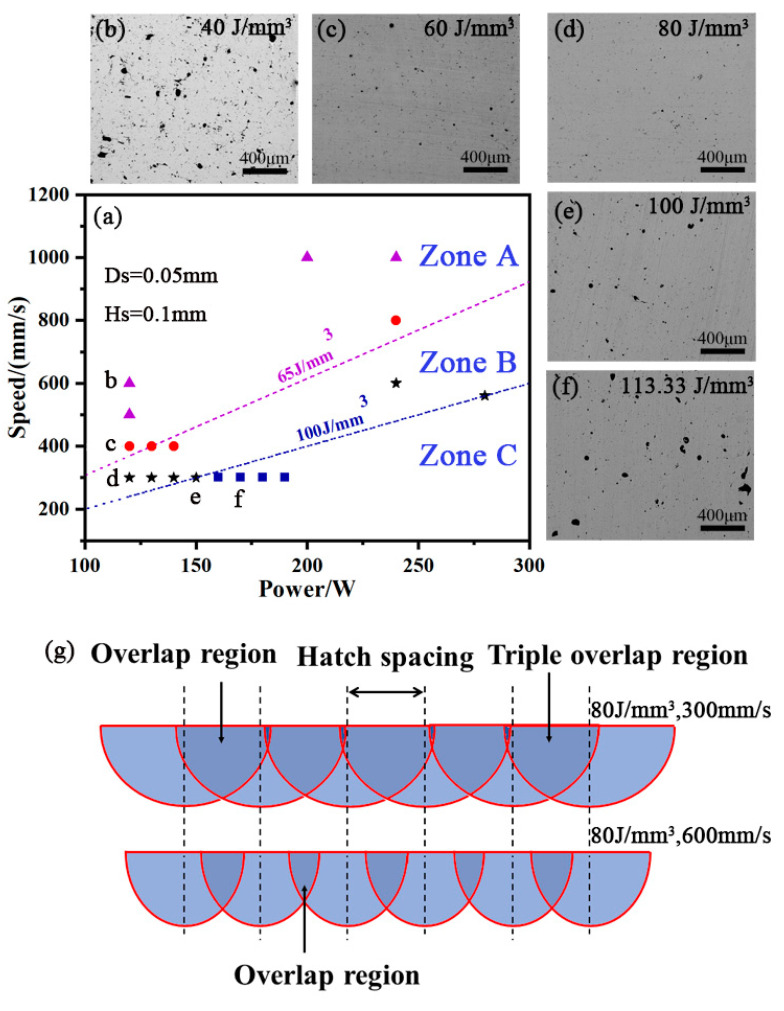
(**a**) Processing map of AS41 samples formed by SLM with different laser powers and scanning speeds, (**b**–**f**) microstructure of samples at different Ev, (**g**) schematic illustration of the effect of the same Ev with different scanning speeds on the molten pool morphology.

**Figure 4 materials-15-05863-f004:**
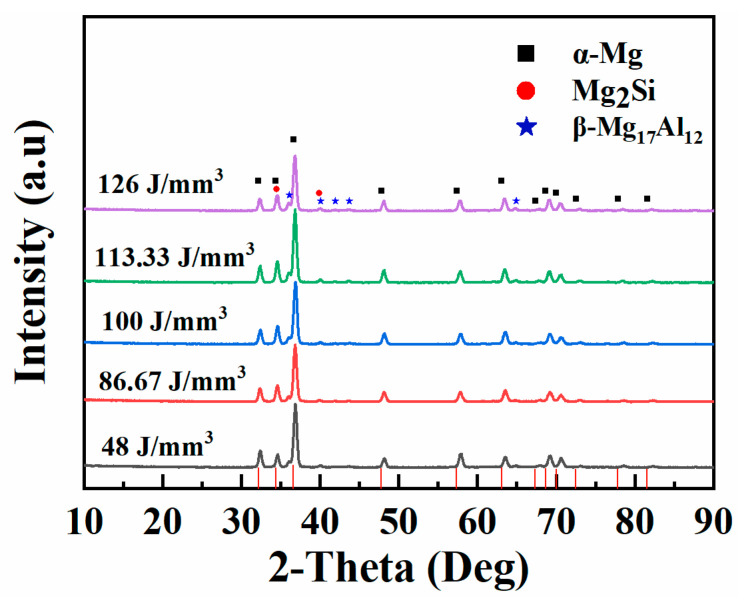
XRD patterns of the SLM samples at different Ev.

**Figure 5 materials-15-05863-f005:**
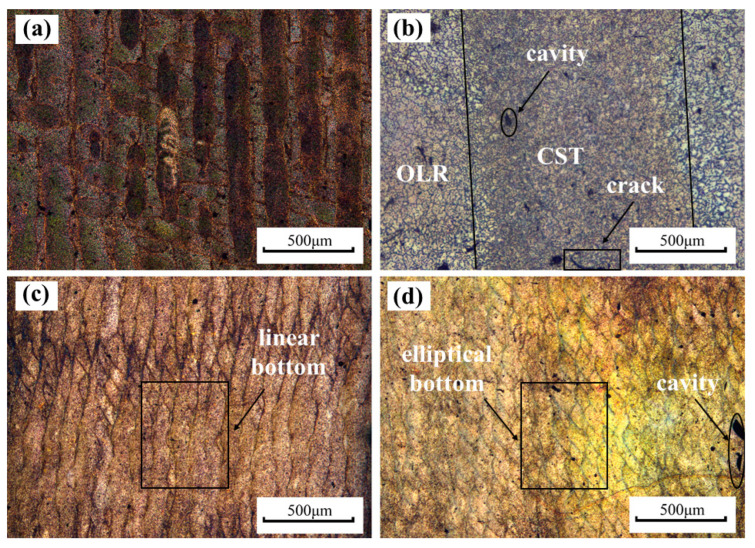
Metallographic microscope photograph of samples at different Ev. The cross-section (**a**) and (**b**) at 80 J/mm^3^, the longitudinal section (**c**) at 80 J/mm^3^ and (**d**) at 65 J/mm^3^.

**Figure 6 materials-15-05863-f006:**
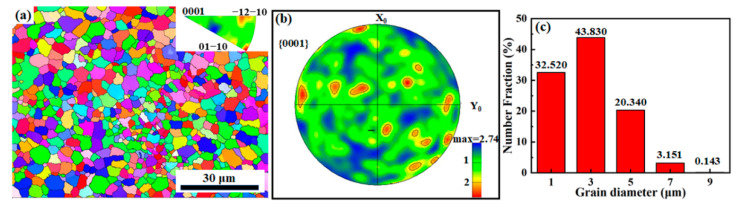
EBSD of AS41 Mg alloy sample along the cross-section: (**a**) the grain orientation and inverse polar map, (**b**) the polar figure, (**c**) the grain size distribution.

**Figure 7 materials-15-05863-f007:**
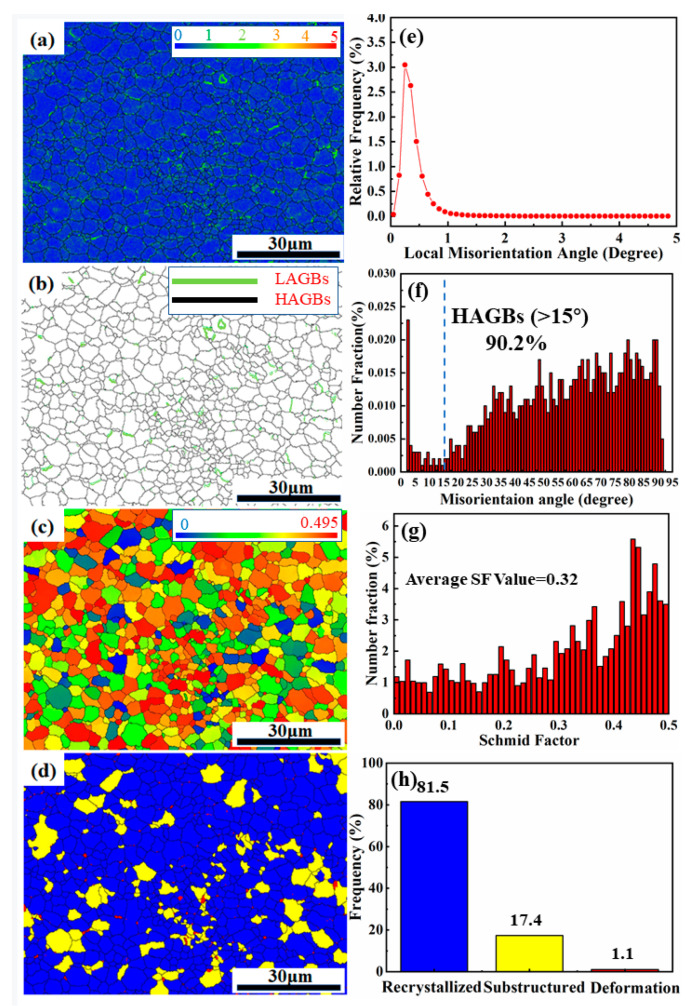
EBSD of AS41 Mg alloy sample along the cross-section: (**a**,**e**) the kernel average misorientation map, (**b**,**f**) grain boundary misorientation map, (**c**,**g**) Schmid factor distribution, (**d**,**h**) the recrystallization distribution.

**Figure 8 materials-15-05863-f008:**
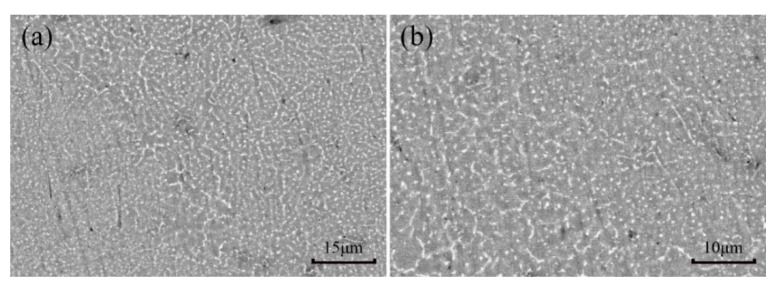
(**a**,**b**) SEM micrographs of AS41 sample at 80 J/mm^3^.

**Figure 9 materials-15-05863-f009:**
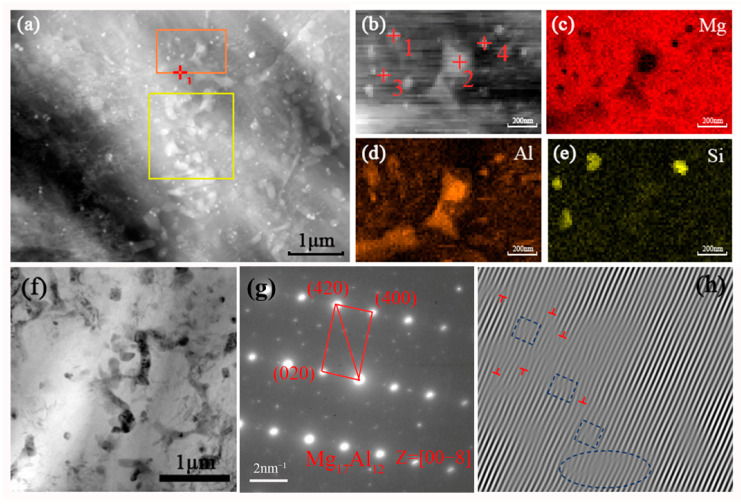
(**a**) STEM image of the second phase in AS41 sample, (**b**–**e**) chemical mapping, (**f**) TEM micrograph, (**g**) SAED pattern, (**h**) inverse Fourier filtered transform pattern.

**Figure 10 materials-15-05863-f010:**
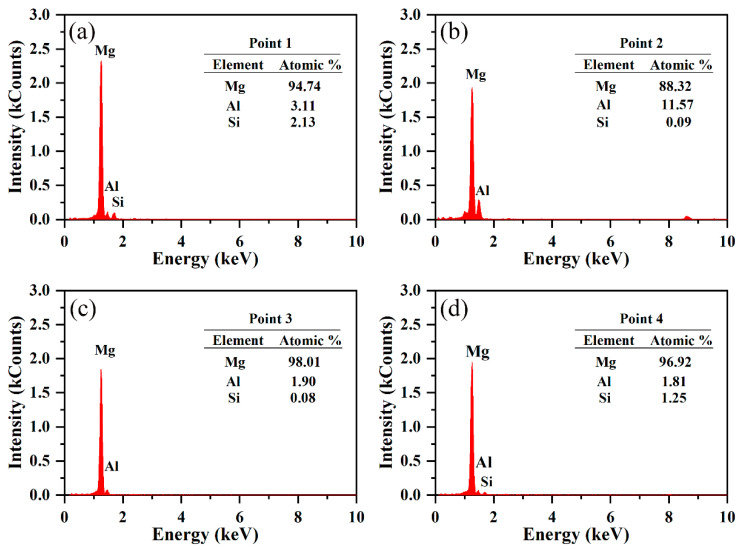
(**a**–**d**) EDS analysis at point 1, 2, 3, 4 in Figure 9b.

**Figure 11 materials-15-05863-f011:**
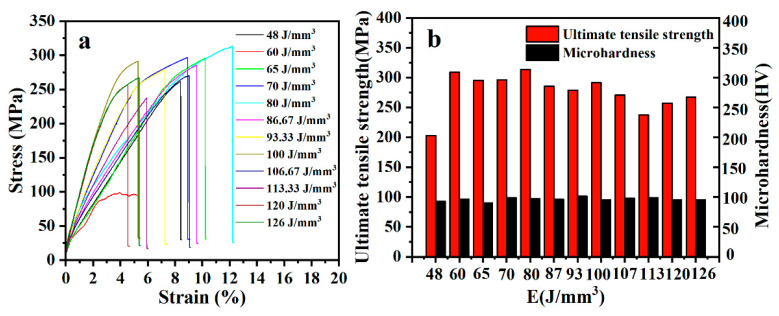
(**a**) Stress-strain curves, (**b**) UTS and microhardness of AS41 samples formed by SLM at different laser energy densities.

**Figure 12 materials-15-05863-f012:**
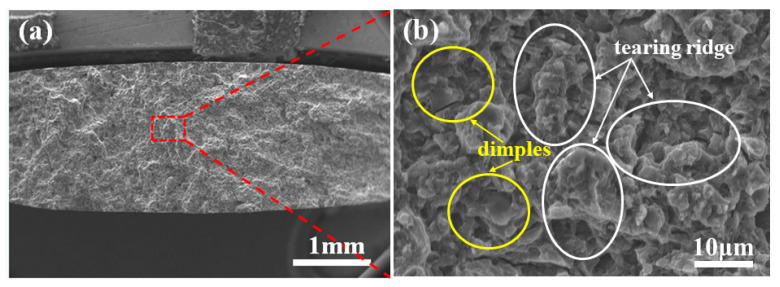
Fracture morphologies of SLM tensile samples: (**a**) the overall fracture surface morphology, (**b**) the local magnification of fracture surface morphology.

**Table 1 materials-15-05863-t001:** Chemical compositions of AS41 powders (wt.%).

	Mg	Al	Si	Zn	Mn
**AS41**	93.49%	4.29%	1.42%	0.2%	0.19%

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
