# Peer review of "Enhanced Strength and Hardness of AS41 Magnesium Alloy Fabricated by Selective Laser Melting"

_materials, 2022, doi:10.3390/ma15175863_

Round 1

Reviewer 1 Report

The manuscript presents a detailed study on the fabrication of AS41 Mg alloy (Mg alloy with Al and Si) by selective laser melting (SLM), aimed at improving considerably the alloy's mechanical properties,  compared to conventional casting technology. The formation of a high-performance AS41 alloy with smaller average grain size and exhibiting weak texture and fine-grain strengthening effects is achieved by optimising the volume laser density power. By characterizing and investigating the microstructure and mechanical properties of the alloy a significant improvement in the metallurgical quality of the SLM fabricated alloy is demonstrated. As a result of the whole research on the mechanism of SLM formation of AS41 Mg alloy theoretical and technical references for the preparation of high-performance Mg alloy are provided.

The reported results are novel, informative and of significance to researchers in the field. The paper is well structured and the overall quality of the contribution is good.

However, the authors could improve the manuscript by providing some information about the used laser device as wavelength, pulse duration and repetition rate, beam-spot dimensions, and scanning speed, which are of practical importance to the researchers working in this field.

Other comments:

·        Fig 2 – the dimension units are missing. (mm ?)

·        Fig.3 a - the insets in the figure are overlapped, unclear and not informative.

·        For clearness, the A, B and C zones and corresponding volume energy density should be defined either in the text or better in the corresponding figures (Fig.3 b-d).

·        Fig. 3g is also not very clear and may arouse different interpretations (for example, the laser spot dimensions change with changing scanning speed ?).

·        In Fig.7 b the light green colour is hardly visible.

·        There are no comments in the text for Fig.10 .

·        Line 401 - Is something missing or it is just a typo?

Reviewer 2 Report

·       At first, the work was very good, and I recommended it be published after minor revisions.  

·       English language should be polished.

·       The introduction section, I feel, can be enhanced by adding an extra paragraph that compares the previous work and SLM methods for improving the mechanical and microstructure properties depending on the new phases formed under different fabrication processes.

·       The novelty of the work should be mentioned in the last paragraph; hence, the authors only mention what they do. You should demonstrate the difference between the current work and the previous literature.

·       In figure 2, the authors should mention that all dimensions in mm

·       In figure 7, e,f,g and h have low resolutions. I suggest the authors add a clear image with corresponding x-y axis titles.

·       In figure 8 , the authors mention that the image demonstrates a massive nanoscale, but the image is on a microscale. Please clarify or replace it with nanoscale.

·       Page 14 line 401 “ Error! Reference source not found.” Please revise the contains

·       In Figure 12. -b  the red color “tearing ridge”  please change into white text color to be clear for the reader

Reviewer 3 Report

The submitted manuscript entitled ‘Enhanced strength and hardness of AS41 magnesium alloy fabricated by selective laser melting’ is delaing with the production of macrosamples by selective laser melting of AS41 magnesium alloy. The main contribution in the submission is the dependence of basic mechanical properties and microstructure on the energy input. During the review of the manuscript, the following issues arose.

Main concerns:

1.      The properties of the cast version of the same alloy should be reported as comparison. Please also discuss the additional cost: SLM versus casting. Is it worth to additively produce such parts instead of casting?

2.      AS41 alloy is hardanable by precipitation hardening. Why were the samples not heat treated?

Additional issues:

1.      Please add the provider of the Mg alloy powder.

2.      How was the chemical composition listed in table 1 measured?

3.      What is the reason behind the exactly 67° degree between the scanning paths?

4.      Please add the sequence and details of the sample preparation (grinding, polishing, etc.).

5.      Fig 2: please do not use ‘*’ to indicate multiplication, please use ‘×’ (not ‘x’!) instead.

6.      Fig 3a: the labelling of the micrographs are incorrect, it should start by ‘(b)’.

7.      Fig 3b-3f: are the black spots pores? If so, the porosity of the samples should be reported.

8.      Fig 5: the labels are unreadable, please select different colour.

9.      Fig 8: please indicate the microstructural phases, features.

10.  Mechanical properties: please report all the standardized results of the tensile test, not only the UTS: proof strength, elongation at fracture, area reduction, etc. please refer to the corresponding ISO standard.

11.  Is fig 12a a fracruture surface? If so, why is it lenticular?

Reviewer 4 Report

The paper “Enhanced strength and hardness of AS41 magnesium alloy fabricated by selective laser melting” presents selective laser melting (SLM) technique fabricating AS41 samples and explored the effect of laser energy densities on the metallurgical quality. From my point of view, the article is of great interest and its publication would have great impact and interest, only a few minor changes could improve the holistic vision of the topic:

·        The position of Table 1 is inadequate.

·        Explain better the battery of tests carried out and porosity analysis, like in:

o   https://doi.org/10.1016/j.jmatprotec.2021.117271

o   https://doi.org/10.1016/j.addma.2020.101663

·        Could you please indicate the standard used to carry out the tensile tests, if you have followed one?

·        Please change any verbs used in the personal form "we..." to the impersonal form.

·        How the samples were attacked to reveal the grain in the specimens for microstructure analysis.

·        The labels in Figure 5 do not read very well and worse in the printed version, if you can change them.

·        In the conclusions, please introduce some quantitative values

These are small suggestions for paper publication.

Round 2

Reviewer 3 Report

Thank you for all the changes and corrections, in the opinion of this Reviewer, the manuscript is now ready for publication.

However, the final decision belongs to the Editor of course.